# OpenReview forum: "Mapping the Digital Terrain: An Insight to Ethical Hacking"
_IEEE.org/ICIST/2024/Conference — IEEE ICIST 2024 Conference Submission_

### Official Review · Reviewer_K5pA · 2024-08-29
**This guide is designed to offer a comprehensive list of resources that are important for those exploring the realm of eth ical hacking.**

**Rating:** 6
**Confidence:** 5

**Review:**

1.This article resembles more of an introductory manual, it is recommended that authors summarize the contribution of the method proposed in this article in the introduction of the article to highlight the research advantages of this article.
2.Ethical hacking is a critical component of modern cybersecurity, but there are several ongoing issues and challenges in the field that need to be addressed, such as Legal and Regulatory Compliance, Scope Creep, Data Privacy, and Ethical Hacking in Emerging Technologies. Addressing these issues requires ongoing collaboration between ethical hackers, organizations, and regulatory bodies to ensure that ethical hacking remains an effective and responsible practice in the ever-evolving landscape of cybersecurity. How do the authors consider this problem?
3.The expression is unclear; for example, when referring to a figure, instead of saying "see 4.1," it should be revised to "see the figure 4.1."
4.The formatting is incorrect. It is recommended to adjust the alignment in item 6 of the Implementation section.

---

### Official Review · Reviewer_tyfF · 2024-08-29
**Accept**

**Rating:** 6
**Confidence:** 3

**Review:**

This paper focuses on ethical hacking, which is a critical technology in modern cybersecurity. Overall, the paper presents an important study that which ethical hackers play an indispensable role in strengthening digital defences by proactively identifying and addressing security vulnerabilities before they can be exploited by malicious actors. However, several aspects could benefit from clarification, revision, or additional analysis.
1. The writing could be more concise and clearer in some sections.
2. Ensure all claims are properly supported by citations, and check for missing or outdated references.
3. The discussion of limitations and future work could be expanded to more fields.

---

### Official Review · Reviewer_xxtg · 2024-08-30
**The reviewer is not familiar with the research content of this article**

**Rating:** 7
**Confidence:** 2

**Review:**

Are there distinct sections or recommendations tailored to varying levels of expertise within the realm of ethical hacking?

---

### Decision · Program_Chairs · 2024-09-06

Accept (Oral)